# Low WT1 Expression Identifies a Subset of Acute Myeloid Leukemia with a Distinct Genotype

**DOI:** 10.3390/cancers17071213

**Published:** 2025-04-03

**Authors:** Michela Rondoni, Giovanni Marconi, Annalisa Nicoletti, Barbara Giannini, Elisa Zuffa, Maria Benedetta Giannini, Annamaria Mianulli, Marianna Norata, Federica Monaco, Irene Zaccheo, Serena Rocchi, Beatrice Anna Zannetti, Adele Santoni, Claudio Graziano, Monica Bocchia, Francesco Lanza

**Affiliations:** 1UO Ematologia, Ospedale S. Maria delle Croci, Via Randi 5, 48121 Ravenna, Italy; giovanni.marconi@unibo.it (G.M.); francesco.lanza@auslromagna.it (F.L.); 2Department of Medicine and Surgery (DIMEC), University of Bologna, 40126 Bologna, Italy; 3U.O. Genetica Medica, AUSL della Romagna, Piazzale della Liberazione 60, 47522 Pievesestina di Cesena, Italy; 4IRCSS Istituto Romagnolo per lo Studio dei Tumori “Dino Amadori”—IRST S.r.l., 47014 Meldola, Italy; 5UO Ematologia, Ospedale Infermi, Viale Luigi Settembrini 2, 47923 Rimini, Italy; 6Dipartimento Scienze Mediche, Chirurgiche e Neuroscienze, University of Siena, 53100 Siena, Italy

**Keywords:** *WT1*, AML, low *WT1* expression, CHIP mutations, marrow dysplasia, measurable residual disease, prognosis, next-generation sequencing

## Abstract

Acute myeloid leukemia is a type of blood cancer that develops due to genetic changes in bone marrow cells. One gene, *WT1*, is usually highly expressed in AML and is commonly used as a disease marker. However, little is known about cases where WT1 expression is unusually low at diagnosis. In this study, we analyzed the genetic characteristics and clinical outcomes of AML patients with low *WT1* expression. We found that these patients often have multiple genetic mutations associated with clonal hematopoiesis and marrow dysplasia, conditions linked to a more complex and slowly evolving form of AML. This suggests that low *WT1* expression may be a marker of a distinct disease subtype.

## 1. Introduction

Acute myeloid leukemia (AML) is a heterogeneous hematologic malignancy characterized by the clonal proliferation of myeloid precursors [1]. Wilms’ tumor gene 1 (*WT1*), found on chromosome 11p13, was initially identified as a tumor suppressor gene implicated in Wilms’ tumor but has since been recognized for its role in leukemogenesis. WT1 is highly expressed in most AML cases, and its role as a biomarker implicated in both prognosis and disease progression has been hypothesized in the initial studies regarding this gene [2,3,4]. Some studies demonstrated that the aberrant persistence of WT1 overexpression after induction chemotherapy and during the subsequent courses is associated with an increased risk of relapse, making it a significant marker for disease monitoring and risk stratification in AML patients for therapeutic response [5,6,7,8,9].

More recently, *WT1* has been suggested as a critical biomarker for AML, not only due to its high expression in leukemic cells but also because of the relationship between WT1 expression and recurrent cytogenetic mutations which are known to drive AML pathogenesis [10,11]. The involvement of *WT1* in maintaining leukemia cell proliferation, as shown in experimental models, underscores its importance as a therapeutic target [12,13,14,15,16]. As more than 90% of AML patients have WT1 hyperexpression at diagnosis, WT1 expression was used as an MRD marker [6,7,8,17,18].

Moreover, molecular studies have revealed that *WT1* mutations, though infrequent, are associated with a poor prognosis in AML. These mutations often coexist with other genetic alterations such as those affecting *TET2* or *IDH1/2*, which further complicates the prognostic landscape. Studies have shown that *WT1* mutations disrupt key cellular processes such as DNA methylation, contributing to leukemic progression [19,20].

No data are reported on the relationship between *WT1* status and secondary-type mutations, known to play a role in the identification of secondary AML. The molecular landscape of AML represents the basis for the therapeutic decisional process at diagnosis today, and the latest European Leukemia Net guidelines are based mainly on the identification of molecular subgroups and also on the large application of NGS. The high-risk AML group is increasing with the identification of seven new gene mutations associated with secondary AML, and these “new” mutations have a prevalent role in splicing and epigenetic regulation. Secondary AML defines a subset of the disease with notoriously adverse outcomes. Based on preceding myelodysplastic neoplasms (MDS), myeloproliferative neoplasms (MPNs), or therapy-related clonal aberrations, secondary AML is associated with lower remission rates and overall survival compared with de novo AML [18,19,20,21].

Given the multifaceted role of *WT1* in AML, it is crucial to further explore its mechanistic contributions to leukemogenesis and its potential utility as a biomarker for both prognosis and therapeutic decision-making. This study specifically addresses the population of patients with low expression levels of *WT1*, which were not considered in most of the MRD-oriented studies conducted up to now. Thus, we aim to further elucidate the role of *WT1* expression in AML prognosis.

## 2. Methods

### 2.1. Study Design and Patient Population

This retrospective cohort study was conducted at three institutions within our comprehensive cancer and research network. We included patients diagnosed with AML according to WHO 4th edition [22], who presented with low *WT1* expression at diagnosis. *WT1* levels were quantified using real-time quantitative PCR (qPCR), and a WT1/ABL1 ratio below 250 was considered indicative of low *WT1* expression. All included patients were diagnosed between 2013 and 2017 and had available bone marrow DNA samples stored at our central laboratory. In total, 34 patients were included in the analysis. Clinical data on diagnosis, treatment, and outcome were retrieved from health clinical records for all the accessible patients.

### 2.2. WT1 Expression

Total RNA was extracted from the mononucleated cells of a BM (bone marrow) aspirate and PB (peripheral blood) samples using a Maxwell CSC RNA blood kit (Promega, Madison, WI, USA); cDNA was generated by a commercial kit based on standardized EAC retrotranscription (REF). The quantification of the *WT1* transcript was performed using RT-PCR with a *WT1* profile Quant (ELN) Ipsogen kit (Qiagen, Hilden, DE, USA). Absolute quantification was determined, and WT1 overexpression was defined as ≥50 copies WT1/ABL1 10^4^ in PB samples and ≥250 copies WT1/ ABL 10^4^ in BM samples [23]; consequently, samples with *WT1* hypoexpression considered for this study were defined as BM samples with WT1/ABL1 10^4^ < 250.

### 2.3. Next-Generation Sequencing (NGS)

We performed NGS using Sophia myeloid solution^®^ (Sophia Genetics SA, Rolle, CH, USA) on bone marrow DNA samples for the targeted sequencing of 32 genes commonly mutated in AML to assess the mutational landscape of each patient. Libraries were prepared according to manufacturer protocols, and sequencing was conducted on an Illumina platform with a minimum depth of 500× coverage (Illumina Inc, San Diego, CA, USA). Mutations were called using established bioinformatics pipelines, and variants were classified as either pathogenic, likely pathogenic, or of uncertain significance based on existing databases (COSMIC, ClinVar) and the literature. Variants with a variant allele frequency (VAF) ≥5% were considered significant.

### 2.4. CHIP and MR Mutation Detection

To evaluate the presence of clonal hematopoiesis of indeterminate potential (CHIP)-related mutations and mutations related to marrow dysplasia (MR), we evaluated within our panel the presence of *DNMT3A*, *TET2*, and *ASXL1* for CHIP mutations and *ASXL1*, *EZH2*, *RUNX1*, *SF3B1*, *SRSF2*, *U2AF1*, and *ZRSR2* as MR genes according to ICC22 (*BCOR* and *STAG2* were not covered by our panel) [24].

### 2.5. Statistical Analysis

The incidence of mutations in the cohort was compared with published data on AML mutational profiles [25]. Differences in the frequency of specific mutations between the low-*WT1* cohort and a general AML population were assessed using Fisher’s exact test or chi-squared tests, as proper. Survival analysis was performed using Kaplan–Meier methods, and comparisons between groups were made using a log-rank test. Statistics were calculated with R-4.4.1 [26].

### 2.6. Dealing with Missing Data

Our study was primarily based on laboratory data. In our cohort, we collected data on sex, age, date of diagnosis, and date of sampling. This small data-set was the only set available for our patients. Furthermore, survival follow-up data were available for 26/32 patients.

### 2.7. Ethical Considerations

This study was approved by the institutional review boards of all participating institutions. Given the retrospective nature of this study and the use of de-identified samples, the requirement for informed consent was waived. All procedures were conducted in accordance with the Declaration of Helsinki.

## 3. Results

### 3.1. Low-Level WT1 Is Associated with CHIP and MR Mutations

We retrieved frozen bone marrow DNA samples for all patients that presented at diagnosis with a WT1/ABL1 level lower than 250 at the three institutions of our comprehensive cancer and research network from 2013 to 2017. This study included a cohort of 34 patients.

When we performed the NGS characterization of 32 genes, we discovered an incidence of 3.4 mutations per patient, significantly higher than the standard mutation rate described at diagnosis for AML. Through NGS analysis, we saw a notable prevalence of MR mutations in patients with low *WT1* expression, with 22/34 patients (65%) affected by mutations related to marrow dysplasia. Specifically, mutations in genes such as *SRSF2* and *RUNX1* were often shown (Figure 1A). Furthermore, we noted that CHIP(DTA) mutations were particularly common in our set, affecting most of the patient population (23/34, 67%, Figure 1A). Within *FLT3*- or *NPM1*-mutant patients, all patients except one had MR or CHIP(DTA) co-mutations; the sole patient without any of these had a *WT1* mutation. Finally, *TP53* mutations were frequent, being present in all six patients affected without any co-mutations, seeming to be the sole driver of their disease. Most of the mutations were missense, frameshift, or nonsense, as expected in loss-of-function mutations (Figure 1B). For all the CHIP and MR mutations, the variant allele fraction was consistent with non-passenger mutations (Figure 1C); see also Appendix A.

Among the patients analyzed, we hypothesized that harboring low-level WT1 expression may be a hallmark of complex disease, with mutations in key myeloid tumor suppressor genes accumulating over time, signifying a significant correlation between low *WT1* expression and adverse genomic profiles. The presence of multiple (≥3) concurrent mutations was seen in 25/34 patients, further emphasizing the association between low *WT1* expression and complex mutational status.

### 3.2. The Incidence of MR Mutations Is Higher than Expected in a Standard Population of AML

To confirm our hypothesis, we compared the incidence of mutations in our low-level WT1 cohort against a larger AML population as reported [25]. The focus was on figuring out whether the incidence of specific mutations in our cohort was significantly higher or lower than expected based on the larger population. We detected a higher incidence of *TET2, ASXL1*, *RUNX1*, *EZH2*, *ZRSR2*, and *SRSF2* mutations. *FLT3* was significantly less mutated in the low-level WT1 cohort. A trend toward an increased incidence was seen for *TP53* mutations, detected in 6/34 patients (18%), while a trend toward a decreased incidence was seen for *NPM1* mutations. Mutations in *NRAS*, *IDH2*, *CEBPA*, *ZRSR2*, and *PTPN11* showed no statistically significant differences between our cohort and the Papaemmanuil cohort [25], despite variations in frequencies. *U2AF1*, *KRAS*, *KIT*, and *CBL* mutations were seen at low frequencies in both cohorts, with no significant differences detected (Table 1). A comparison shows that certain MR mutations, notably in genes involved in chromatin and DNA structure, are significantly more frequent in the low-level *WT1* cohort compared to the other AML population. This suggests that patients with low *WT1* expression might have a distinct mutational profile, potentially influencing their clinical outcomes and therapeutic responses.

### 3.3. Low-Level WT1 Impacted Patients’ Prognosis

Twenty-six patients were evaluable for survival. These patients had a median age at diagnosis of 70 years (IQR 64.9, 75.8), and 15/26 patients were male. The median survival time was 327 days (Figure 2), and the probability of being alive 3 years after diagnosis was 28%. Treatment was administered according to the standard of care, with intensive treatments being administered for 8/26 patients and non-intensive treatment administered for 18/26 patients. Clinical characteristics were available for these 26 patients and are reported in Table 2; most of the patients harbored the hallmark of complex cytogenetics and had secondary disease. The survival results were not comparable with more extensive sets, due to the rarity and granularity of our population. Furthermore, our population mimicked the biological/genetic characteristics of MR AML or TP53 AML; thus, due to the number of confounding factors and the rarity of the phenomenon, the low expression of *WT1* was not treated in our study as a candidate for a negative prognostic factor.

## 4. Discussion

Our study sheds light on the complex relationship between *WT1* expression and the mutational landscape in AML. We found that low-level *WT1* expression is associated with a higher incidence of multiple concurrent mutations per patient, particularly those linked to CHIP and marrow dysplasia, as defined in the recent ICC classification [24]. Even if our population was primarily diagnosed according to the WHO2016 classification, we discovered that the CHIP ICC 22 category may be particularly present in our patients. Although antecedent clinically relevant myelodysplasia may have also been encountered in some of our patients, since we are not able to access the anamnestic data of patients, this is one of the main limitations of our study. These findings suggest that low *WT1* expression may serve as a surrogate marker for an underlying mutational complexity, standing for a unique disease biology that evolves over time.

A notable finding of this study is the prevalence of mutations in key DTA genes, namely *DNMT3A*, *TET2*, and *ASXL1*, in patients with low *WT1* expression. This supports the notion that WT1 downregulation in AML might reflect the accumulation of mutations characteristic of CHIP and thereafter MR-AML, which often signify a more slow-growing but biologically complex form of the disease. Consistently, MR mutations are also enriched in our study population. These mutations have been previously described as driver mutations in the pathogenesis of AML and are known to adversely affect prognosis, particularly in elderly patients [27,28,29]. The presence of MR mutations, which are frequently involved in dysplastic processes via the negative regulation of gene expression via chromosomal CpG island methylation and chromatin organization, points to a disease biology driven by differentiation and maturation defects with a subverted transcriptional program and nuclear structure [30,31,32]. In our experience, all 34 cases of AML with low *WT1* expression, except 1, harbored at least a CHIP or MR mutation. Significantly, this also applies to patients with classical de novo mutations, such as *NPM1* and *FLT3*. The sole exception, a case without any CHIP or MR mutations, harbored a *WT1* mutation, suggesting that *WT1* itself may play a role in driving leukemogenesis in the absence of other high-risk mutations [19]. This is in line with earlier studies showing that *WT1* mutations take part in deregulating transcriptional programs critical for myeloid differentiation and proliferation [9,14].

Likewise, the high number of mutations seen in patients with low *WT1* expression aligns with the hypothesis that low *WT1* expression may be indicative of a disease that has evolved over a long time through the accumulation of genetic lesions and remain slow-growing. If confirmed in a larger set, this may be significant in terms of therapeutic decision-making, as these mutations often predict resistance to standard therapies and may guide the use of targeted or experimental treatments [19,24,33]. With the prevalence of *TP53* mutations and MR mutations treated in a pre-venetoclax era, when the option of bone marrow transplant was reserved for younger patients, and in a population that mainly received non-curative treatments, survival was not evaluable in our set, and this is another limitation of our study.

The role of *WT1* in the biology of AML extends beyond its function as a simple response marker. In our cohort, patients with low *WT1* expression showed a distinct mutational signature, including a higher frequency of mutations associated with adverse clinical outcomes. This suggests that WT1 expression is not merely a passive consequence of leukemogenesis but may actively contribute to the malignant process by promoting a specific set of mutations that confer a selective advantage to the leukemic clone [14,19]. In our set, where *WT1* expression did not contribute to leukemogenesis, we are able to depict a biologically different evolution.

In conclusion, our study highlights the biological significance of *WT1* expression in AML. Low *WT1* expression appears to be associated with a biologically distinct subset of AML characterized by multiple high-risk mutations, particularly CHIP and MR mutations. These findings underscore the need for further research into the role of *WT1* in AML pathogenesis, with a view toward refining prognostic models and developing tailored treatment strategies for patients with low *WT1* expression levels. As our understanding of *WT1’s* role in AML continues to evolve, it may become a key biomarker for guiding therapeutic decisions and improving patient outcomes.

## 5. Conclusions

Our study highlights the biological and clinical significance of low *WT1* expression in AML, demonstrating that it is associated with a distinct mutational profile enriched in CHIP and MR mutations. These findings suggest that AML patients with low *WT1* expression may represent a biologically unique subgroup, potentially linked to secondary AML or disease evolution over time. The increased frequency of mutations in genes involved in chromatin modification and RNA splicing further supports the hypothesis that low *WT1* expression reflects an accumulation of genetic lesions rather than a simple biomarker variation.

## Figures and Tables

**Figure 1 cancers-17-01213-f001:**
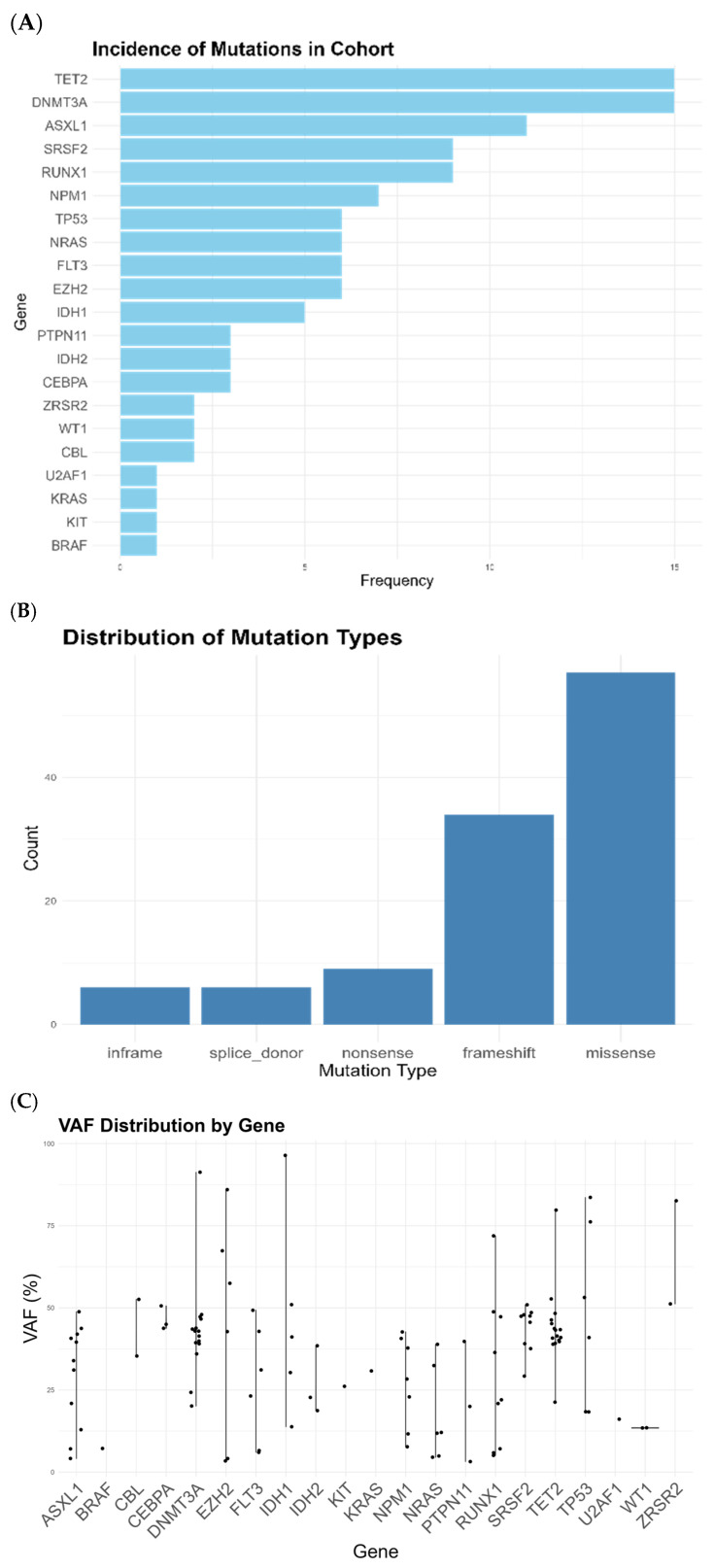
(**A**) Incidence of gene mutations in cohort of 29 patients with low-level WT1 expression. (**B**) Mutation per type. (**C**) Median variant allele fraction for mutations in each gene.

**Figure 2 cancers-17-01213-f002:**
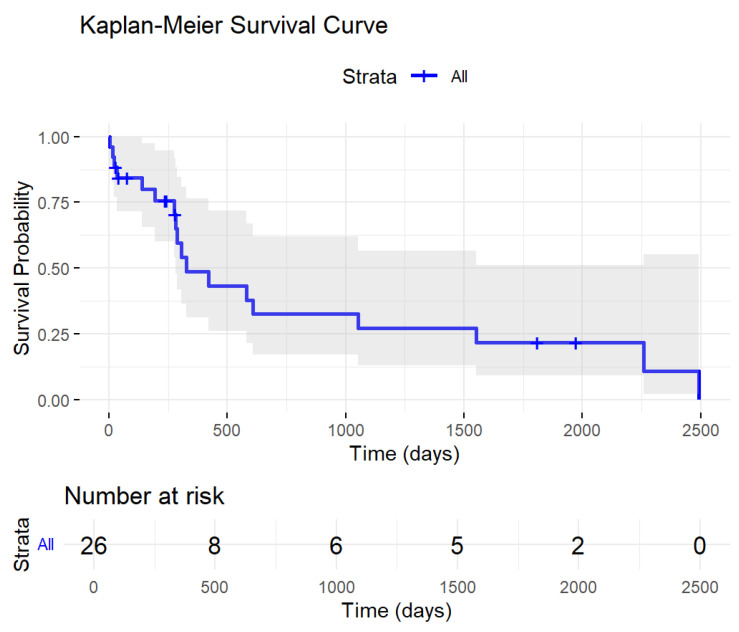
Kaplan-Meier survival of 26 patients from our laboratory’s low-level WT1 cohort.

**Table 1 cancers-17-01213-t001:** A comparison between the incidence of mutations in the low-level WT1 cohort and data retrieved from the literature.

Gene	Low-Level WT1 Cohort	Papaemanuouil et al. [25]	*p* Value
*DNMT3A*	13 (14%)	382 (15.9%)	0.222
*TET2*	11 (11.8%)	205 (8.5%)	0.017
*ASXL1*	10 (10.8%)	71 (3.0%)	0.001
*RUNX1*	9 (9.7%)	151 (6.3%)	0.013
*TP53*	6 (6.5%)	110 (4.6%)	0.057
*SRSF2*	7 (6.5%)	93 (3.9%)	0.030
*FLT3*	8 (6.5%)	572 (23.8%)	0.017
*EZH2*	9 (6.5%)	48 (2.0%)	0.001
*NPM1*	5 (5.4%)	438 (18.2%)	0.068
*NRAS*	3 (3.2%)	292 (12.1%)	0.160
*IDH2*	3 (3.2%)	152 (6.3%)	1.000
*CEBPA*	3 (3.2%)	216 (9.0%)	0.490
*ZRSR2*	2 (2.2%)	13 (0.5%)	0.048
*PTPN11*	2 (2.2%)	131 (5.5%)	0.770
*IDH1*	2 (2.2%)	105 (4.4%)	1.000
*U2AF1*	1 (1.1%)	38 (1.6%)	0.612
*KRAS*	1 (1.1%)	80 (3.3%)	1.000
*KIT*	1 (1.1%)	71 (3.0%)	1.000
*CBL*	1 (1.1%)	42 (1.7%)	1.000

**Table 2 cancers-17-01213-t002:** Clinical characteristics of 26 patients with available data.

**Characteristic**	**Patients (n = 26)**
**Age, n (IQR)**	70 (64.9–75.8)
**Sex (M/F)**	15/11
**Karyotype**	
- Complex - −5 - −7 - del(17p) - Normal - Other alterations - Not evaluable/not available	3/26 2/26 3/26 1/26 3/26 3/26 11/26
**Secondary to myelodysplasia, n(%)**	11/15 (75%) [11 not known]

## Data Availability

Anonymized data are included in the Appendix A. Data may also be requested in different formats addressing a specific request to the corresponding author.

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
