# Peer review of "Low WT1 Expression Identifies a Subset of Acute Myeloid Leukemia with a Distinct Genotype"

_cancers, 2025, doi:10.3390/cancers17071213_

Round 1
Reviewer 1 Report
Comments and Suggestions for Authors
1. Please unify the term minimal residual disease (MRD) in Introduction section and measurable residual disease (MRD) in the Abstract.
2. Please include the reference for quantifying WT1levels using real-time quantitative PCR (qPCR).
3. Separate paragraph for line 209 " In conclusion, our study...."
Author Response
Dear Reviewer,
We enormously thank you for your positive comments.
A systematic response to any comment follow:
- Please unify the term minimal residual disease (MRD) in Introduction section and measurable residual disease (MRD) in the Abstract.
Term was unified (changes: abstract)
- Please include the reference for quantifying WT1levels using real-time quantitative PCR (qPCR).
A chapter on WT1 level quantification was added (changes: lines 77-82, ref 22)
- Separate paragraph for line 209 " In conclusion, our study...."
Paragraph was separated

Reviewer 2 Report
Comments and Suggestions for Authors
In this article the authors analysed 34 AML patients with low WT1 expression and used NGS to identify mutations in 32 genes frequently mutated in AML. The frequency of mutations found was compared with published data on AML mutational profiles. It was concluded that low WT1expression in AML was associated with a distinct and complex mutational profile linked with CHIP and myelodysplastic related (MR) mutations.
This manuscript contains some points that must be reviewed:
- As the patients with low WT1 expression presented mutations, particularly in ASXL1, TET2, and SRSF2 related with clonal hematopoiesis (CHIP) or myelodysplasia mutations, it is necessary to review if these patients presented a previous myelodysplastic syndrome.
The authors may include the subgroups of AML that was classified the 34 patients studied and the descritpion of the dysplasias and the karyotype of bone marrow cells. It was not clear the type of AML studied, if it was de novo AML, or secondary AML. The authors cited the article “Marconi G, Rondoni M, Zannetti BA, et al. Novel insights and therapeutic approaches in secondary AML. Front Oncol. 2024” in the discussion of the manuscript. So, it is necessary in the introduction of this paper be clear the diference between de novo AML and secondary AML, and also include in the methods the AML subgroups according to WHO classification.
-It is necessary to review the genes nomenclature, the genes must be write in italic. This revision is necessary in the manuscript.
-In the introduction, lines 41-44, the authors may replace “The prognostic significance of WT1 overexpression, however, remains controversial, with some studies linking high WT1 expression to poor outcomes, while others suggest it may be a marker of minimal residual disease (MRD) and therapeutic response” for “Some studies demonstrated that the prognostic significance of WT1 overexpression is associated with poor outcomes , and it may be a marker of minimal residual disease (MRD) and therapeutic response.” According to the abstract of the manuscript.
- In the methods, lines 67-75, it may be included the demographic and clinical charcteristics of the 34 patients studied, these data may be presented in a Table with, age, sex, subgroup of AML classification, dysplasias in the bone marrow, karyotype, percentage of blasts. An in the results, lines it is not necessary the phrase154-156 “Due to the nature of our laboratory cohort, full clinical data was not available. Sufficient clinical data was available in 26 patients. These patients had a median age at diagnosis of 70 years (IQR 64.9, 75.8), and 15/26 patients were male”. In the methods, it may be included the description of the WT1 expression analyses using the real time quantitative PCR, since it is the main theme of the manuscript.
- In the results, it is recommended to described first the results of the WT1 expression. What was used as controls of the experiments, healthy donors of bone marrow? It was not clear if this analysis was performed using a higher number of patients and then it was selected only the patients who showed a low WT1 expression.
-In the discussion the authors cited the ICC classification, reference 22, and used for the classification of the patients the WHO 4th edition, reference 21. It is necessary to review this point. It may be clear if these patients had a previous MDS to review some points in the discussion. It may be included the possible cause of the presence of dysplastic cells in the patients studied. In the final of the discussion, before the conclusion, the authors may include the limitations of this study.
- It is recommended to review the resolution of figure 1 and the legend of figure 2 “A) KM survival of 26 patients from our laboratory low-level WT1 cohort”.
Author Response
Dear reviewers,
We enormously thank you for your positive comments.
A systematic response to any comment follow:
- As the patients with low WT1 expression presented mutations, particularly in ASXL1, TET2, and SRSF2 related with clonal hematopoiesis (CHIP) or myelodysplasia mutations, it is necessary to review if these patients presented a previous myelodysplastic syndrome. The authors may include the subgroups of AML that was classified the 34 patients studied and the descritpion of the dysplasias and the karyotype of bone marrow cells. It was not clear the type of AML studied, if it was de novo AML, or secondary AML. The authors cited the article “Marconi G, Rondoni M, Zannetti BA, et al. Novel insights and therapeutic approaches in secondary AML. Front Oncol. 2024” in the discussion of the manuscript. So, it is necessary in the introduction of this paper be clear the diference between de novo AML and secondary AML, and also include in the methods the AML subgroups according to WHO classification.
We reviewed the cohort, and we included a table with general characteristics of our clinical patients set (table 2). Thus, we confirmed that most patients had secondary leukemia and harbored complex cytogenetics
- It is necessary to review the genes nomenclature, the genes must be write in italic. This revision is necessary in the manuscript.
Genes nomenclature was reviewed
- In the introduction, lines 41-44, the authors may replace “The prognostic significance of WT1 overexpression, however, remains controversial, with some studies linking high WT1 expression to poor outcomes, while others suggest it may be a marker of minimal residual disease (MRD) and therapeutic response” for “Some studies demonstrated that the prognostic significance of WT1 overexpression is associated with poor outcomes , and it may be a marker of minimal residual disease (MRD) and therapeutic response.” According to the abstract of the manuscript.
We accepted this suggestion and made the change
- In the methods, lines 67-75, it may be included the demographic and clinical charcteristics of the 34 patients studied, these data may be presented in a Table with, age, sex, subgroup of AML classification, dysplasias in the bone marrow, karyotype, percentage of blasts. An in the results, lines
We reviewed the cohort, and we included a table with general characteristics of our clinical patients set (table 2). Thus, we confirmed that most patients had secondary leukemia and harbored complex cytogenetics
- In the results, it is recommended to described first the results of the WT1 expression. What was used as controls of the experiments, healthy donors of bone marrow? It was not clear if this analysis was performed using a higher number of patients and then it was selected only the patients who showed a low WT1 expression.
A chapter on WT1 level quantification was added (changes: lines 77-82, ref 22)
- In the discussion the authors cited the ICC classification, reference 22, and used for the classification of the patients the WHO 4th edition, reference 21. It is necessary to review this point. It may be clear if these patients had a previous MDS to review some points in the discussion. It may be included the possible cause of the presence of dysplastic cells in the patients studied. In the final of the discussion, before the conclusion, the authors may include the limitations of this study.
We tried to clarify that WHO16 was used for diagnosis, while ICC 22 is only an argument for the discussion. At the end, our study seed lights on the pathogenesis of complex AMLs.
- It is recommended to review the resolution of figure 1 and the legend of figure 2 “A) KM survival of 26 patients from our laboratory low-level WT1 cohort”.
We re-drawn the figures with a better font.
Reviewer 3 Report
Comments and Suggestions for Authors
Thank you for inviting me to review this manuscript. Here, the authors examine the mutational burden and its potential consequences on survival in a group of AML patients with low expression of WT1. Although the cohort is small to make general conclusions, I believe that considering that the disease is rare and this particular subtype is ever more rare, this work is important and deserves to be published following some corrections:
1) Please, improve the readability of the figure by increasing the font sizes and removing the grids. Keep font sizes consistent
2) Proofread the text as there are some language shortcomings and typos.
3) Include more details about PCR detection of WT1 including probe sequences, material used and qPCR conditions. Show the data in the supplementary materials including relevant quality controls for qPCR.
4) If the authors have mutational profile for the patients with high WT1 expression from their center, it would be interesting to compare the mutational burden with them. If that data is not available, is it possible to subgroup Papaemmanuil et al. cohort by WT1 expression at all?
5) When the authors discuss the survival data, they should provide an explanation of what they mean by “surprisingly good survival” by comparing it to the general AML population as they did for the mutational profile. Name confounding factors in the discussion considering that the number of patients in the survival analysis decreased even further.
6) Comment in the discussion on the fact that higher number of mutations does not negatively impact the survival in this study. The authors state that high mutational burden in low expression WT1 patients may influence therapeutic decisions as it may be indicative of a disease which developed during a long time; how does this go together with the fact that these patients had “surprisingly good survival” under standard treatment?
Comments on the Quality of English LanguagePay attention to consistency in the text and in figure legends.
Author Response
Dear reviewers,
We enormously thank you for your positive comments.
A systematic response to any comment follow:
- Please, improve the readability of the figure by increasing the font sizes and removing the grids. Keep font sizes consistent
We re-drawn the figures with a better font.
- Proofread the text as there are some language shortcomings and typos.
The article was carefully proofreaded
- Include more details about PCR detection of WT1 including probe sequences, material used and qPCR conditions. Show the data in the supplementary materials including relevant quality controls for qPCR.
A chapter on WT1 level quantification was added (changes: lines 77-82, ref 22)
- If the authors have mutational profile for the patients with high WT1 expression from their center, it would be interesting to compare the mutational burden with them. If that data is not available, is it possible to subgroup Papaemmanuil et al. cohort by WT1 expression at all?
At the moment, we prefer not to publish these data. In our centers, NGS was performed extensively in a more recent patient set (2021 on). Patients studied in early 2018 were only complex cases. Thus, a preliminary analysis we conducted confirms our finding with minor differences, however, patient populations are markedly different in time and characteristic and comparison is not worthy. Also, we overtly risk underestimating some signals due to low numbers in our center compared with them, or to capture false signals due to selection bias. Finally, our number are not high, and statistical power may be lsot (approximately 220 cases WT1 high in our set vs 1500 in literature). This Is a population of consecutive patients. Similar populations of consecutive patients in literature provides a better, and more reliable comparison for this informative analysis. No change.
- When the authors discuss the survival data, they should provide an explanation of what they mean by “surprisingly good survival” by comparing it to the general AML population as they did for the mutational profile. Name confounding factors in the discussion considering that the number of patients in the survival analysis decreased even further.
- Comment in the discussion on the fact that higher number of mutations does not negatively impact the survival in this study. The authors state that high mutational burden in low expression WT1 patients may influence therapeutic decisions as it may be indicative of a disease which developed during a long time; how does this go together with the fact that these patients had “surprisingly good survival” under standard treatment?
We agree with this point. We decided to put a more cautious sentences in results and discussion (changes: lines 171-175, 215 – 219) we also revise discussion precising that no prognostic and treatment decision may be derived from our data. (lines 211-212)
Round 2
Reviewer 2 Report
Comments and Suggestions for Authors
This second version of the manuscript entitled “Low WT1 expression identifies a subset of AML with distinct genotype” still presents some points that are necessary to review carefully. For example, in methods, page 2, in the WT1 expression section, the authors mentioned "(REF)", but did not add the reference. Therefore, it is necessary to review all the references. Also in this section, the authors indicated how the analysis of high expression was done, but it was not indicated how the analysis of low expression was defined. This is important because it is the focus of the study. In table 2, it is recommended to replace “17p del” for "del(17p)". In table 2, it is not mentioned which is the second disease. It is recommended to review the English, for example on page 3, line 111, “.... date of diagnisis...” . In the supplementary material there are some parts that are not in English. The table of the supplementary data is not recommended to be included in the article, since it presents the names of the patients. The number of patients studied is limited.
Comments on the Quality of English LanguageIt is recommended an English revision.
Author Response
Comment 1: This second version of the manuscript entitled “Low WT1 expression identifies a subset of AML with distinct genotype” still presents some points that are necessary to review carefully. For example, in methods, page 2, in the WT1 expression section, the authors mentioned "(REF)", but did not add the reference. Therefore, it is necessary to review all the references.
Reply: references have been added
Also in this section, the authors indicated how the analysis of high expression was done, but it was not indicated how the analysis of low expression was defined. This is important because it is the focus of the study.
Reply: definition has been addes
In table 2, it is recommended to replace “17p del” for "del(17p)".
Reply: edited
In table 2, it is not mentioned which is the second disease.
Reply: edited
It is recommended to review the English, for example on page 3, line 111, “.... date of diagnisis...” .
reply: edited
In the supplementary material there are some parts that are not in English.
Reply: edited
The table of the supplementary data is not recommended to be included in the article, since it presents the names of the patients.
Reply: edited and included in supplementary
The number of patients studied is limited.
Reply: this is a limit of the study
Reviewer 3 Report
Comments and Suggestions for Authors
Dear authors,
Thank you so much for trying to make adjustments according to my comments. I accept that you would not like to disclose the mutational burden of your cohort. I have some minor comments left:
-line 75-85 contains a typo the heading of that section and the (REF) in the text which should probably be substituted with a real reference.
-line 109-112: please, proofread. Multiple typos.
-check figure 1 again. I think the adjustments led to some movements of the text, make sure it does not propagate to the publication
-please, style table 2 the same way as other table and proofread! Typos seem to be an issue
Author Response
Reviewer 2: Thank you so much for trying to make adjustments according to my comments. I accept that you would not like to disclose the mutational burden of your cohort. I have some minor comments left:
-line 75-85 contains a typo the heading of that section and the (REF) in the text which should probably be substituted with a real reference.
Reply: added and edited
-line 109-112: please, proofread. Multiple typos.
Reply: edited
-check figure 1 again. I think the adjustments led to some movements of the text, make sure it does not propagate to the publication
Reply: We double checked and re-positioned the B letter
-please, style table 2 the same way as other table and proofread! Typos seem to be an issue
Reply: We formatted the table and proofreaded.
